# Mediating-Moderating Effect of Allostatic Load on the Association between Dietary Approaches to Stop Hypertension Diet and All-Cause and Cause-Specific Mortality: 2001–2010 National Health and Nutrition Examination Surveys

**DOI:** 10.3390/nu11102311

**Published:** 2019-09-29

**Authors:** Hind A. Beydoun, Shuyan Huang, May A. Beydoun, Sharmin Hossain, Alan B. Zonderman

**Affiliations:** 1Department of Research Programs, Fort Belvoir Community Hospital, Fort Belvoir, VA 22060, USA; 2Department of Anesthesiology, Fort Belvoir Community Hospital, Fort Belvoir, VA 22060, USA; 3Laboratory of Epidemiology and Population Sciences, National Institute on Aging, NIA/NIH/IRP, Baltimore, MD 21225, USA

**Keywords:** allostatic load, diet, hypertension, mortality, survey

## Abstract

This secondary analysis of survey data examined mediating-moderating effects of allostatic load score (calculated using the Rodriquez method) on the association between nutrient-based Dietary Approaches to Stop Hypertension (DASH) diet score (Mellen Index) and the all-cause and cause-specific mortality risks among 11,630 adults ≥ 30 years of age from the 2001–2010 National Health and Nutrition Examination Surveys with no history of cardiovascular disease or cancer at baseline, and who were followed-up for ~9.35 years. Multivariable models were adjusted for demographic, socioeconomic, lifestyle, and health characteristics. All-cause, cardiovascular disease, and cancer-specific mortality rates were estimated at 6.5%, 1.1%, and 1.9%, respectively. The median DASH total score was 3.0 (range: 1–8) (with 78.3% scoring < 4.5), whereas the median allostatic load score was 3 (range: 0–9). The DASH diet, fiber, and magnesium were negatively correlated with allostatic load, whereas allostatic load predicted higher all-cause mortality, irrespective of the DASH diet. Whereas protein was protective, potassium increased all-cause mortality risk, irrespective of allostatic load. Potassium was protective against cardiovascular disease-specific mortality but was a risk factor for cancer-specific mortality. Although no moderating effects were observed, mediation by the allostatic load on cardiovascular disease-specific mortality was observed for DASH total score and selected component scores. Direct (but not indirect) effects of DASH through the allostatic load were observed for all-cause mortality, and no direct or indirect effects were observed for cancer-specific mortality. From a public health standpoint, the allostatic load may be a surrogate for the preventive effects of the DASH diet and its components on cardiovascular disease-specific mortality risk.

## 1. Introduction

Although previous studies have emphasized the role played by single nutrients in chronic disease prevention, more recent studies have concentrated on dietary patterns which take into account complex interactions among nutrients [1]. Current evidence supports the detrimental health impact of Western dietary patterns that are high in animal products, soft drinks, and processed foods, as well as the beneficial health impact of alternative dietary patterns that are high in vegetables, fruits and fiber-rich cereals [2]. Healthy dietary patterns that are consistent with the Healthy Eating Index, the Mediterranean-style, and the Dietary Approaches to Stop Hypertension (DASH) have been linked to a reduced risk of chronic diseases, including the two leading causes of death, namely cardiovascular disease and cancer [2,3,4,5,6,7,8,9]. A recently published update on a systematic review and meta-analysis of prospective cohort studies has established that healthy dietary patterns could reduce the risks of all-cause mortality, cardiovascular disease, cancer, type 2 diabetes, and neurodegenerative diseases by 22%, 22%, 16%, 18%, and 15%, respectively [3].

Dietary patterns may also influence cardiometabolic risk factors, including obesity, hypertension, hyperglycemia, and hyperlipidemia [10], as well as markers of atherosclerosis and inflammation [11]. Hypertension, defined as persistent diastolic blood pressure (DBP) ≥90 mmHg and systolic blood pressure (SBP) ≥140 mmHg, is prevalent among 20–35% of the global population of adults and has been associated with an increased likelihood of cardiovascular disease, end-stage renal disease, and mortality [8]. Past research has established a diet and other lifestyle factors as major contributors to the onset of hypertension [12]. Accordingly, non-pharmacological strategies for the prevention and treatment of hypertension have emerged, including the DASH dietary pattern [12,13].

Hypertension remains the most frequently diagnosed health problem in the United States [14], and the DASH diet was initially developed in the 1990s to target this modifiable risk factor for chronic disease development [2,4,10,15]. DASH scoring systems were developed using targets for specific food groups [1,2,4,5,6,7,11,13,15,16] and nutrients [14,16,17]. The ability of the DASH dietary pattern to reduce blood pressure and prevent hypertension among pre-hypertensive individuals was demonstrated in randomized controlled trials [2,15]. Subsequent epidemiologic studies concluded that the DASH dietary pattern might reduce cardiovascular disease and type 2 diabetes risks [4], as well as the all-cause mortality in selected populations, including adults with hypertension, women with heart failure and individuals >60 years of age [7,11,13,14]. To date, a limited number of studies have examined the DASH diet in relation to all-cause and cause-specific mortality risks using nationally representative samples of the U.S. population [4,14]. Furthermore, few studies have attempted to evaluate the mediation-moderation of these relationships by specific factors in relatively healthy populations [10].

The development of chronic conditions, including cardiovascular disease and cancer, has been attributed to age, genetics, diet, smoking, physical activity, as well as environmental stressors. These factors may interact and potentially lead to poor health through various biopsychosocial mechanisms. A state of chronic stress may result from such interactions, and the impact of chronic stress has been repeatedly operationalized as an allostatic load [18,19]. 

Broadly speaking, the DASH dietary pattern has been linked to reductions of 8–22% in all-cause mortality, 19–28% in cardiovascular disease-specific mortality and 11–23% in cancer-specific mortality risks [9]. Given that the DASH dietary pattern was initially developed to prevent the onset of hypertension, biopsychosocial mechanisms linking the DASH dietary pattern to all-cause, and cause-specific mortality risks require further elucidation. Since diet can be viewed as an environmental stressor that may influence an individual’s cardiometabolic profile, it is plausible that the link between a DASH dietary pattern and the mortality risk may be mediated or moderated by a wear-and-tear phenomenon which occurs when different physiological systems react to environmental stressors, namely the allostatic load [18,19]. The purpose of this secondary analysis of survey data was to examine the mediating-moderating effect of the allostatic load score [18]–defined using selected indicators of cardiometabolic risk, glucose metabolism, cardiopulmonary functioning, parasympathetic functioning, and inflammation—on the association between the DASH dietary pattern and all-cause, cardiovascular disease and cancer-specific mortality risks among adults ≥30 years of age who participated in the 2001–2010 National Health and Nutrition Examination Surveys (NHANES) and had no history of cardiovascular disease or cancer at baseline.

## 2. Materials and Methods

### 2.1. Database

This study was conducted in accordance with the Declaration of Helsinki as revised in 2013. We performed secondary analyses of an existing, de-identified, and public-use database that was generated from combining multiple waves from the NHANES. Initiated by the Centers for Disease Control and Prevention (CDC) National Center for Health Statistics (NCHS), the NHANES is a series of nationally representative surveys that used multi-stage, stratified sampling designs to assess the health and nutritional status of civilian, non-institutionalized adults and children in the United States. Details of the CDC/NCHS/NHANES design, methodology, and procedures are provided on the following website: https://www.cdc.gov/nchs/nhanes/index.htm. Demographic, socioeconomic, dietary, and health data were collected by trained staff in a mobile examination center (MEC) or during in-home visits. Anthropometric, physiological, and laboratory measurements were collected for all, or in some cases, a sub-group of study participants. The original NHANES protocol was approved by the NCHS Institutional Review Board, with written informed consent obtained from all study participants, whereas the current study was determined as being research not involving human subjects [20]. For these analyses, we combined the 2001–2002, 2003–2004, 2005–2006, 2007–2008, and 2009–2010 NHANES waves whereby the C-reactive protein (CRP) level, a marker of inflammation and a key component of allostatic load, was assessed, and we subsequently applied a series of selection criteria to fulfill the study purpose.

### 2.2. Study Sample

A total of 52,195 participants from the 2001–2010 NHANES were enumerated in the initial sample. Of those, 22,650 (43.4%) were adults ≥ 30 years of age. After excluding participants with a history of cardiovascular disease and cancer, a total of 17,571 (77.6%) adults ≥30 years remained in the study sample. Furthermore, we sequentially excluded participants on the basis of the following criteria: CRP ≥10 mg/dL (n = 1642); extreme levels of energy consumption (<400 kilocalories or >5000 kilocalories) (n = 948); died within the first year of follow-up (n = 134); missing data on main exposure variables (DASH score and its components, allostatic load score) (n = 3217). The resulting study sample consisted of 11,630 (66.2%) participants, of whom 2525 (21.7%) had missing data on the covariates, and 9105 (78.3%) had no missing data on covariates and was accordingly labeled as the complete subject database. The exclusion of NHANES participants with CRP > 10 mg/dL was intended to reduce confounders related to the presence of inflammatory diseases that may affect dietary patterns, as well as mortality risks. Assuming data were missing at random and using non-missing data on age, sex, and race, multiple imputations (k = 10) were performed for 2525 with missing data on specific covariates (education, marital status, smoking status, alcohol consumption, physical activity, self-rated health, BMI), resulting in a final analytic sample of 11,630 participants (Figure 1). Mediation analyses were based on the complete subject database, whereas the remaining analyses were based on the imputed database. 

### 2.3. Measures

#### 2.3.1. DASH Dietary Pattern

The 2001–2010 NHANES included a dietary history component whereby a 24-hour recall was administered by trained interviewers in the MEC, and dietary intake data were collected using computerized techniques. Subsequently, nutrient intakes were estimated by linking dietary intake with corresponding U.S. Department of Agriculture nutrient databases [17].

The DASH dietary pattern was designed to provide 6–8 servings of grains, 4–5 servings of vegetables, 4–5 servings of fruit, 2–3 servings of fat-free or low-fat milk products, ≤6 ounces of meat products, 4–5 servings of nuts, seeds, or legumes, 2–3 servings of fats and oils, and <5 servings of sweets and added sugars each week [16]. The majority of similarly conducted studies [1,2,4,5,6,7,11,13,15,16] have operationalized the DASH dietary pattern by calculating a DASH score based on consumption of eight food categories, as proposed by Fung and colleagues [21], namely, fruits, vegetables, nuts and legumes, low-fat dairy, whole grains, sodium, sweetened beverages, as well as red and processed meats. Specifically, participants in the highest quintile of fruits, vegetables, nuts and legumes, low-fat dairy, and whole grains received a score of 5, and those in the lowest quintile received a score of 1, whereas participants in the highest quintile of sodium, sweetened beverages, and red and processed meats received a score of 1 and those in the lowest quintile received a score of 5 [21]. Participants in the intermediate quintiles received intermediate scores. The score for each component was summed to get an overall DASH score that ranges between 8 and 40 [1,2,4,5,6,7,11,13,15,16,21]. 

Since the purpose of this study was to elucidate the biopsychosocial mechanism underlying the diet-mortality link and for consistency with previously conducted studies using NHANES, we operationalized the DASH dietary pattern using a methodology developed by Mellen and colleagues [17], which is solely based on nutrients and has been previously utilized in the context of 24-hour dietary recall data collected by NHANES [4,14,16,17]. A total DASH score or Mellen index that ranges between 0 and 9 was computed based on absolute targets for nine nutritional components (total fat, saturated fat, protein, fiber, cholesterol, calcium, magnesium, potassium, sodium) while assuming a 2100 kilocalories diet for both sexes. For each of the nine DASH components, study participants who met the specified goal received 1 point, whereas those who met an intermediate goal received 0.5 points and those who met neither goal received 0 points. Furthermore, we dichotomized the total DASH score such that participants who met approximately half of the DASH targets (score ≥ 4.5) were considered consistent with the DASH dietary pattern [4,14,16,17] (Appendix A).

#### 2.3.2. Allostatic Load

Selected variables from the 2001–2010 NHANES examination and laboratory modules were used to evaluate allostatic load. Allostatic load has been previously conceptualized as the outcome of cumulative effects of repeated or chronic exposure to chemical and non-chemical stressors that result in a shift from a normal to an adaptive but dysfunctional state which can negatively impact physical and mental health [18,19]. Previously, Thomson and colleagues [19] described an Allostatic Load Index (ALI) as a composite score ranging between 0 and 9 using thresholds or percentiles of selected biomarkers (total cholesterol, high-density lipoprotein (HDL), glycated hemoglobin (HbA1c), waist-to-hip ratio, SBP, DBP, resting heart rate, CRP, and serum albumin), with higher scores indicating worse allostatic load. The 2001-2010 waves of NHANES data lacked the anthropometric measurement of hip circumference; therefore, we adopted the definition of ALI as described by Rodiqeuz and colleagues [18]. Specifically, high-risk (coded as 1), moderate-risk (coded as 0.5), and low-risk (coded as 0) categories were generated for the following nine allostatic load components: SBP: ≥150 mmHg [1], 120 to <150 mmHg [0.5], <120 mmHg [0]; DBP: ≥90 mmHg [1], 80 to <90 mmHg [0.5], <80 mmHg [0]; body mass index (BMI): ≥30 kg/m^2^ [1], 25 to <30 kg/m^2^ [0.5], <25 kg/m^2^ [0]; HbA1c: ≥6.5% [1], 5.7% to <6.5% [0.5], <5.7% [0]; total cholesterol: ≥240 mg/dL [1], 200 to <240 mg/dL [0.5], <200 mg/dL [0]; HDL cholesterol: <40 mg/dL [1], 40 to <60 mg/dL [0.5], ≥60 mg/dL [0]; total/HDL cholesterol ratio: ≥6 [1], 5 to <6 [0.5], <5 [0]; CRP: ≥3 mg/L [1], 1 to <3 mg/L [0.5], <1 mg/L [0]; albumin: <3.0 μg/mL [1], 3.0 to <3.8 μg/mL [0.5], ≥3.8 μg/mL [0]; and creatinine clearance: <30 mL/min/1.73 m^2^ [1], 30 to <60 mL/min/1.73 m^2^ [0.5], ≥60 mL/min/1.73 m^2^ [0]. A total score ranging between 0 and 9 was obtained for the ALI by adding unweighted component scores (Appendix A). 

#### 2.3.3. All-Cause and Cause-Specific Mortality

The primary outcome was all-cause mortality rate, and secondary outcomes were cardiovascular disease and cancer-specific mortality rates. Vital status and person-months of follow-up were obtained from the linkage between baseline NHANES databases and the National Death Index death certificate data. To date, mortality follow-up is complete until 31 December 2015. The ICD-10 codes for the underlying causes of death variable were used to define cause-specific mortality rates. Specifically, cardiovascular deaths included diseases of the heart (I00–I09, I11–I13, and I20–I51) as well as cerebrovascular diseases (I6–I69), whereas cancer deaths included malignant neoplasms (C00–C97). 

#### 2.3.4. Covariates

Demographic and socioeconomic characteristics included age (in years), sex (Male, Female), race (Mexican American, Other Hispanic, non-Hispanic White, non-Hispanic Black, Other), education (less than high school, 9–11th Grade, High School Graduate/General Education Development or Equivalent, Some College or AA degree, College Graduate or above), marital status (married/living with partner, other) and poverty income ratio [PIR] (<100%, 100%–<200%, ≥200%). Lifestyle characteristics were defined as the smoking status (non-smoker, ex-smoker, current smoker), alcohol consumption [≥12 glasses in the past 12 months] (yes, no), and physical activity [walking/bicycling, tasks around home/yard, moderate activity or vigorous activity in the past 30 days] (yes, no). Finally, health characteristics were defined as weight status based on BMI categories, and self-rated health (SRH). The BMI was calculated as weight (kg) divided by height squared (m^2^) and categorized using to the World Health Organization definition of weight status, namely underweight/normal weight (BMI: <25 kg/m^2^), overweight (BMI: 25–29.9 kg/m^2^), and obese (BMI ≥ 30 kg/m^2^). The SRH was based on a single general health questionnaire item, namely, “Would you say your health, in general, is excellent, very good, good, fair or poor?” and further categorized as “excellent/very good/good” versus “fair/poor”. 

### 2.4. Statistical Analysis

All analyses were conducted using STATA v. 15 (StataCorp, College Station, TX) and the 2001–2010 NHANES recommended MEC sample weights. Continuous variables were summarized as mean ± standard error of the mean (SEM), whereas proportions were calculated for categorical variables. First, fully-adjusted Cox proportional hazards models were constructed for demographic, socioeconomic, lifestyle, and health characteristics as predictors of all-cause and cause-specific mortality rates. Second, fully-adjusted linear and logistic regression models were constructed for demographic, socioeconomic, lifestyle and health characteristics as predictors of the DASH score, defined as a continuous or categorical (<4.5 vs. ≥ 4.5) outcome. Third, fully-adjusted linear regression models were constructed for demographic, socioeconomic, lifestyle, and health characteristics as predictors of allostatic load score. Fourth, fully-adjusted linear regression models were constructed for DASH score and its component scores as predictors of allostatic load index score. Fifth, we performed Cox proportional hazard regression analyses to evaluate the main and interactive effects of the DASH score (or its component scores) and the ALI score in relation to all-cause and cause-specific mortality rates. All statistics, including means, proportions, SEM, beta coefficients (β), odds ratios (OR), and hazard ratios (HR) with their 95% CI were estimated taking complex sampling design, as well as imputations into consideration. Finally, using the complete subject database and not taking complex sampling design into consideration, we applied the ‘paramed’ method to estimate the controlled direct effect, the natural indirect effect and the total effect (with their 95% CI) that correspond to the paths (DASH diet → allostatic load → mortality), while adjusting for years of follow-up, as well as demographic, socioeconomic, lifestyle, and health characteristics. The ‘paramed’ method performs causal mediation analysis using parametric regression models. Two models are estimated: a model for the mediator conditional on exposure and covariates, and a model for the outcome conditional on exposure, the mediator and covariates. It extends statistical mediation analysis (known as Baron and Kenny procedure) to allow for the presence of treatment (exposure)-mediator interactions in the outcome regression model using counterfactual definitions of direct and indirect effects. The ‘paramed’ method is a user-written STATA package which was based on the MEDIATION macros in the SAS and SPSS that were developed by Valeri and VanderWeele [22]. It allows continuous, binary or count outcomes, and continuous or binary mediators, and requires the user to specify an appropriate form for regression models. These analyses resulted in slope ‘estimates’ with their 95% CI that corresponded to beta coefficients when the outcome was continuous and log OR when the outcome was dichotomous. Two-sided statistical tests were performed at α level of 0.05.

## 3. Results

A total of 11,630 2001–2010 NHANES participants (48.5% male) with mean (±SEM) age of 48.78 (±0.23) years and mean (±SEM) BMI of 28.59 (±0.09) kg/m^2^ were evaluated in this study, with mean (± SEM) follow-up time estimated at 112.27 (±1.11) months or 9.35 (±0.09) years. As shown in Table 1, 72.9% were non-Hispanic White, 82.7% had completed high school education or better, 71.5% were married or living with a partner, 72.8% had PIR ≥200%, 52.9% had never smoked, 65.8% had consumed ≥12 glasses of alcohol in past 12 months, 61.1% were not physically active, and 16.2% had fair or poor SRH. All-cause, cardiovascular disease, and cancer-specific mortality rates were estimated at 6.5%, 1.1%, and 1.9%, respectively. Due to complex sampling design, 10,822 of the 11,630 participants (93.0%) were included in multivariable analyses described in Table 2, Table 3, Table 4 and Table 5. Multivariable analyses using Cox proportional hazard models suggested that older age, male sex as well as fair/poor SRH were positively associated with all-cause and cause-specific mortality rates, whereas “Other” (vs. “Mexican American”) race was positively associated with all-cause but not cause-specific mortality rates. Being married or living with a partner and PIR ≥200% (vs. <100%) were negatively associated with all-cause and cardiovascular disease-specific mortality rates, whereas being an ex-smoker or current smoker was positively associated with all-cause and cancer-specific mortality rates. Finally, non-alcohol drinkers (<12 vs. ≥12 glasses in the past 12 months) were at increased risk of dying from cancer, but not from cardiovascular disease (Table 2).

The mean (±SEM) DASH score was 3.26 (±0.021), and the median (range) DASH score was 3 (1–8), with 78.3% scoring less than 4.5 on the DASH dietary pattern. Similarly, study participants had a mean (±SEM) of 2.82 (±0.02) and a median (range) of 3 (0–9) on the ALI score. Multiple linear and logistic regression analyses suggested that older age and female sex were associated with higher scores whereas higher education, PIR ≥ 200%, ex-smoker or current smoker status, physical inactivity, and greater BMI were associated with lower scores on the DASH dietary pattern. Multiple linear regression analyses suggested that ALI score was positively associated with age, BMI, current smoking, no alcohol consumption, as well as fair/poor SRH, whereas female sex, college graduation, and PIR ≥200% were negatively associated with ALI score (Table 3).

Linear regression models for DASH score and its components as predictors of ALI score are presented in Table 4, before and after adjusting for demographic, socioeconomic, lifestyle, and health-related confounders. In unadjusted models, DASH score and most of the DASH components (except for protein, potassium, and sodium) were significantly and negatively associated with the ALI score. By contrast, DASH score and two DASH components (fiber and magnesium) remained significantly and negatively associated with ALI score, after adjusting for confounders. 

Moderating effects of the ALI on DASH diet-mortality relationships were evaluated in Table 5. The results suggested that ALI score was positively associated with all-cause mortality rate but not a cardiovascular disease or cancer-specific mortality rates, after controlling for confounders. Furthermore, ALI score remained positively associated with all-cause mortality, when the DASH score or its components were entered into the model. By contrast, the DASH score was not significantly associated with mortality rates, either before or after ALI score was entered in the model. There were no statistically significant interaction effects between DASH and ALI scores in relation to all-cause or cause-specific mortality rates, suggesting the absence of moderation by ALI. Similar results were obtained when ALI score was entered along with each of the DASH diet components in multivariable models, with the exception of protein and potassium. Specifically, the DASH protein score was inversely related to all-cause mortality, independently from ALI score and baseline confounders. Also, the DASH potassium score was directly related to all-cause mortality, independently from ALI score and baseline confounders. Interestingly, the DASH potassium score (but not the ALI score) was inversely related to cardiovascular disease-specific and directly related to cancer-specific mortality rates, in fully-adjusted models that include interaction terms. 

Mediating the effects of the ALI on DASH diet-mortality relationships were evaluated in Table 6. The total DASH score and most of the DASH component scores, including saturated fat, protein, fiber, calcium, potassium, and sodium, were directly but not indirectly related to all-cause mortality. Whereas potassium was positively associated, the other DASH scores were negatively associated with all-cause mortality rate. By contrast, the total DASH score and specific DASH component scores (saturated fat, total fat, cholesterol, fiber, magnesium) were indirectly associated with cardiovascular disease-specific mortality rate through the ALI. Finally, there were no direct or indirect relationships between DASH scores and cancer-specific mortality rate. 

## 4. Discussion

In this secondary analysis of survey data, adult NHANES participants with no history of cardiovascular disease or cancer at baseline were followed-up for an average of ~9 years, and inter-relationships among DASH diet, allostatic load, and mortality rates were evaluated. Consistent with past studies [4,5,9,14], all-cause, cardiovascular disease, and cancer-specific mortality rates were estimated at 6.5%, 1.1%, and 1.9%, respectively, and in line with current evidence, the majority of U.S. adults were not adherent to the DASH diet despite recommendations [17]; in fact, 78.3% of study participants reported DASH score <4.5. Also, the mean ALI score was 2.8, consistent with figures reported by Rodriquez and colleagues [18]. Generally speaking, demographic, socioeconomic, lifestyle, and health-related risk factors for mortality and the ALI score were as expected, although negative relationships between socioeconomic indicators (e.g., education, PIR) and the DASH diet necessitate further investigation. 

Whereas DASH total score and two-component scores (fiber and magnesium) were negatively correlated with ALI score, these same dietary factors were not significantly related to all-cause and cause-specific mortality rates, either before or after controlling for ALI score. By contrast, two DASH component scores (protein and potassium) were significantly related to mortality rates, independently of ALI score. Whereas all-cause mortality was negatively related to high protein score and positively related to high potassium score, cardiovascular disease-specific mortality was negatively related, and cancer-specific mortality was positively related to potassium score. In contrast, ALI score was positively related to all-cause but not cause-specific mortality rate, independently of DASH score or its component scores. These findings did not translate into significant moderation effects, although mediation of the DASH-mortality relationship by ALI was observed in the context of cause-specific mortality. Specifically, ALI scores appear to mediate associations of cardiovascular disease-specific but not cancer-specific mortality with total and selected DASH component scores. It is worth noting that many of these nutrients were previously tested as non-pharmacological treatments for hypertension, similarly to the DASH diet [12].

The negative association between the DASH diet and ALI aligns with current epidemiologic research. Studies focusing on female healthcare professionals found an association between the DASH diet and incidence of cardiovascular disease—the top two quintiles of DASH were associated with 36–41% reduced hazards of cardiovascular disease, and multivariate analysis showed 12–23% risk reduction. However, adherence to DASH was not associated with a risk of venous thromboembolism in the same group [10]. In another study evaluating a single versus a combination of dietary nutrients, like DASH, it was shown that the combination diet had greater success in the management of hypertension, as compared to single nutrient. DASH diet with low sodium or with weight loss regimen was as effective as single-drug therapy for the treatment of hypertension [12]. There is evidence of DASH’s cardio-protective effects in the U.K. population as well. Subjects with the most DASH-concordant diets had a 20% reduced incidence of stroke, 13% decreased the total incidence of cardiovascular disease but no change in the risk of coronary heart disease [15]. DASH diet education to stroke patients or patients at risk for stroke was found to be beneficial in another cohort [16]. DASH diet adherence was also found to lower incidence of heart failure by 22% in older Swedish men in the highest DASH quartile [13]. Moreover, adherence to the DASH diet was associated with a lower risk of colorectal cancer in women with a relative risk in the highest quintile of 0.8 for total colorectal cancer and 0.81 for proximal colon cancer [6]. 

The finding that specific DASH components, but not the DASH dietary pattern, may affect the risk of dying from any cause, is consistent with some but not all previously conducted studies. In fact, several studies found a beneficial effect of the DASH diet on all-cause mortality [2] as well as mortality related to multiple chronic diseases like cardiovascular disease, hypertension [8,14], chronic heart failure [13], renal failure [5], stroke [16] and cancer [7,23,24]. For instance, DASH was associated with reduced mortality in healthy middle-aged people, with 20% improvement followed by a reduction in mortality between 8 and 17% [11]. In another study, maintaining a DASH-like diet was associated with reduced risk of total death, gastrointestinal cancers in men, and lower non-gastrointestinal cancers in women [25]. 

It is worth noting that several of the previously conducted studies examined the DASH diet in relation to mortality among individuals with chronic diseases or risk factors for chronic diseases at baseline. In one study, higher DASH scores were associated with a modest decline in mortality in postmenopausal women with heart failure [25]. Adherence to DASH was associated with a 43% lower risk of renal function decline and a 48% reduction in all-cause mortality in adult renal transplant recipients [1]. Furthermore, DASH was associated with lower all-cause mortality in hypertensive adult patients [24], and improved DASH scores were found to be associated with decreased mortality in metabolically obese but normal-weight adults [4]. A high-quality DASH diet was also inversely associated with overall mortality and cancer mortality among cancer survivors [3]. Despite the overwhelmingly significant associations between the DASH diet and mortality, a few studies reported differently. In studies that examined the relationship between DASH score and mortality, no association was found between DASH quality and long-term hypertension or cardiovascular disease mortality in female holders of Iowa driver’s license [26] or decreased cardiovascular or total mortality in heart disease patients, whereby adjusted HR for cardiovascular mortality was 1.19 and HR for all-cause mortality was 1 for highest DASH tertile [5].

To our knowledge, this is the first study to examine the mediation of the relationship between the DASH diet and mortality by allostatic load using the “paramed” method. The observed mediating effects of allostatic load on the relationships between the DASH diet and mortality rates imply that allostatic load may be used as a surrogate measure in future prevention trials involving the DASH diet or its components. From a public health standpoint, prevention of cardiometabolic risk factors that constitute the allostatic load through a dietary modification that is consistent with the DASH diet is likely to reduce the risk of cardiovascular disease-specific deaths, but not cancer-specific deaths. The findings pertaining to specific DASH diet components (especially protein and potassium) require further investigation and replication by others. 

Nevertheless, study findings should be interpreted with caution and in light of several limitations. First, the DASH diet and allostatic load were measured simultaneously, precluding establishment of temporal relationships between these exposures. Similarly, the DASH diet was evaluated using a single 24-hour recall and calculated using nutrients rather than food items. This may have affected observed relationships between the DASH diet, allostatic load, and mortality rates. Third, despite the exclusion of subjects who died within one year of follow-up limiting reverse causality, the maximum follow-up time varied substantially among NHANES participants from distinct waves, potentially affecting the relationship of exposures with the risk of death. Fourth, many of the covariates adjusted for in multivariate analyses can be considered as time-varying, and, in the absence of repeated measurements could only be assessed once in the context of NHANES surveys. Fifth, secondary analyses were performed using existing data which may have limited the ability to ascertain certain characteristics, thus leading to selection or misclassification bias. For instance, a history of cardiovascular disease and cancer were self-reported and could not be ascertained through medical records. Also, data were assumed to be missing at random, and despite multiple imputation the role of selection bias cannot be eliminated. Sixth, the observational nature of this study may have led to residual confounding by unmeasured or inadequately measured confounders. Seventh, sub-samples of the 2001–2010 NHANES participants had valid data on exposures and outcomes of interest, potentially leading to selection bias. Eighth, although a relatively large study sample was analyzed, the role of chance cannot be entirely accounted for in the presence of multiple testing. Of note, with ten exposure variables (DASH total score and nine component scores) and three outcome variables (all-cause, cardiovascular-specific, and cancer-specific mortality rates), similar hypotheses were tested 30 times, potentially reducing alpha to 0.002 per hypothesis after Bonferroni correction. Finally, study findings can only be generalized to U.S. adults, 30 years and older, and future research is needed to confirm study findings in a wider population that includes children, adolescents, and young adults.

In conclusion, the DASH diet, fiber, and magnesium were negatively correlated with allostatic load, whereas allostatic load predicted higher all-cause mortality, irrespective of the DASH diet. Whereas protein was protective, potassium increased the risk of all-cause mortality, irrespective of allostatic load. Furthermore, potassium was protective against cardiovascular disease-specific mortality and a risk factor for cancer-specific mortality. Significant mediation by allostatic load on cardiovascular disease-specific mortality was observed in the context of the DASH total score and selected DASH component scores. Further research is needed to confirm and elucidate these preliminary findings. 

## Figures and Tables

**Figure 1 nutrients-11-02311-f001:**
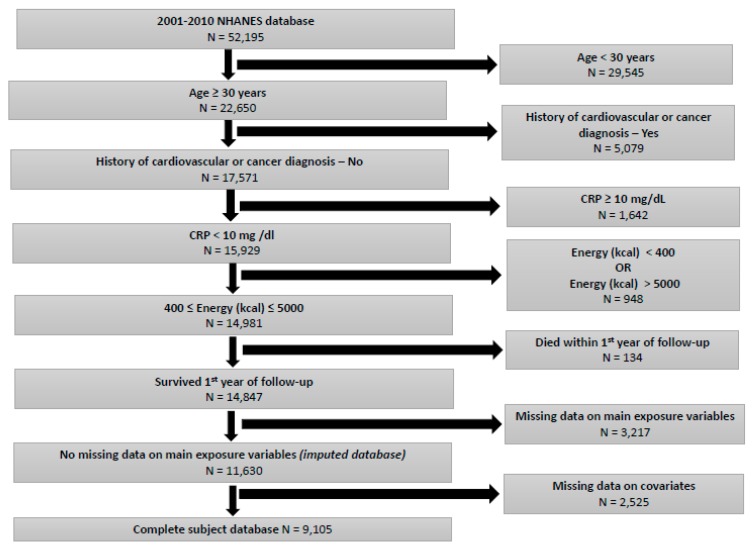
Study Flowchart.

**Table 1 nutrients-11-02311-t001:** Baseline characteristics—2001–2010 NHANES (n = 11,630).

	% or Mean ± SEM (n = 11,630)
**Age (years):**	
Mean ± SEM	48.78 ± 0.23
**Sex:**	
Male	48.5
Female	51.5
**Race/Ethnicity:**	
Mexican American	7.9
Other Hispanic	4.6
Non-Hispanic White	72.9
Non-Hispanic Black	9.5
Other	5.0
**Education:**	
<Less Than 9th Grade	6.3
9–11th Grade	10.9
High School Graduate/General Education Development or Equivalent	24.4
Some College or AA degree	29.4
College Graduate or above	28.9
**Marital status:**	
Married/Living with partner	71.5
Other	28.5
**Poverty-income ratio:**	
<100%	9.9
100%–<200%	17.3
≥200%	72.8
**Smoking status:**	
Never smoker	52.9
Ex-smoker	25.2
Current smoker	21.8
**Alcohol consumption** **(≥12 glasses in the past 12 months):**	
Yes	65.8
No	34.2
**Physical activity:**	
Yes	38.9
No	61.1
**Body mass index (kg/m^2^):**	
Mean ± SEM	28.59 ± 0.09
<25.0	29.8
25.0–29.9	36.1
≥30.0	33.9
**Self-rated health:**	
Excellent/Very good/Good	83.8
Fair/Poor	16.2

Abbreviations: NHANES = National Health and Nutrition Examination Survey, SEM = Standard error of the mean.

**Table 2 nutrients-11-02311-t002:** Cox Proportional Hazards Models for Demographic, Socioeconomic, Lifestyle, and Health Characteristics as Predictors of All-Cause and Cause-Specific Mortality—2001–2010 NHANES (n = 10,822) *.

	All-Cause Mortality (n = 10,822)		Cardiovascular Mortality (n = 10,822)		Cancer Mortality (n = 10,822)	
Age (years):	Log_e_(HR)	95% CI	Log_e_(HR)	95% CI	Log_e_(HR)	95% CI
Mean ± SEM	0.087	0.080, 0.094	0.098	0.083, 0.11	0.072	0.059, 0.085
**Sex:**						
Male	Ref.	--	Ref.	--	Ref.	--
Female	−0.54	−0.71, −0.38	−1.03	−1.43, −0.63	−0.57	−0.98, −0.15
**Race/Ethnicity:**						
Mexican American	Ref.	--	Ref.	--	Ref.	--
Other Hispanic	−0.16	−0.62, 0.29	0.16	−0.65, 0.97	0.15	−0.67, 0.96
Non-Hispanic White	0.031	−0.24, 0.30	0.085	−0.54, 0.71	0.11	−0.38, 0.61
Non-Hispanic Black	−0.068	−0.38, 0.25	0.13	−0.52, 0.79	0.26	−0.34, 0.85
Other	−0.67	−1.22, −0.11	−0.29	−1.64, 1.05	-1.26	−2.54, 0.019
**Education:**						
<Less Than 9th Grade	Ref.	--	Ref.	--	Ref.	--
9–11th Grade	0.21	−0.095, 0.53	0.19	−0.45, 0.84	0.072	−0.62, 0.77
High School Graduate/General Education Development or Equivalent	0.0089	−0.31, 0.33	0.21	−0.39, 0.81	−0.30	−0.99, 0.39
Some College or AA degree	0.12	−0.21, 0.44	0.21	−0.40, 0.82	−0.048	−0.73, 0.63
College Graduate or above	−0.41	−0.82, 0.0091	−0.27	−1.21, 0.66	−0.46	−1.27, 0.35
**Marital status:**						
Married/Living with partner	Ref.	--	Ref.	--	Ref.	--
Other	0.35	0.19, 0.53	0.39	0.0044, 0.79	0.072	−0.28, 0.43
**Poverty-income ratio:**						
<100%	Ref.	--	Ref.	--	Ref.	--
100%–<200%	−0.14	−0.33, 0.058	−0.35	−0.88, 0.18	−0.22	−0.66, 0.22
≥200%	−0.37	−0.59, −0.14	−0.61	−1.18, −0.044	−0.18	−0.66, 0.31
**Smoking status:**						
Never smoker	Ref.	--	Ref.	--	Ref.	--
Ex-smoker	0.29	0.080, 0.52	0.015	−0.47, 0.49	0.53	0.13, 0.92
Current smoker	0.76	0.51, 1.00	0.21	−0.32, 0.73	1.04	0.64, 1.43
**Alcohol consumption** **(≥ 12 glasses in past 12 months):**						
Yes	Ref.	--	Ref.	--	Ref.	--
No	0.12	−0.068, 0.32	0.12	−0.35, 0.60	0.42	0.023, 0.82
**Physical activity:**						
Yes	Ref.	--	Ref.	--	Ref.	--
No	0.059	−0.19, 0.31	0.067	−0.46, 0.60	−0.14	−0.54, 0.26
**Body mass index (kg/m^2^):**						
Mean ± SEM	0.0088	−0.0045, 0.022	0.0043	−0.029, 0.038	0.021	−0.0043, 0.047
<25.0	--	--	--	--	--	--
25.0–29.9	--	--	--	--	--	--
≥30.0	--	--	--	--	--	--
**Self-rated health:**						
Excellent/Very good/Good	Ref.	--	Ref.	--	Ref.	--
Fair/Poor	0.57	0.40, 0.75	0.81	0.40, 1.22	0.42	0.052, 0.79

* Fully-adjusted model. Abbreviations: CI = confidence interval, HR = hazard ratio, NHANES = National Health and Nutrition Examination Survey, SEM = standard error of the mean.

**Table 3 nutrients-11-02311-t003:** Dietary Approaches to Stop Hypertension Diet Score and Allostatic Load Index Score by Demographic, Socioeconomic, Lifestyle, and Health Characteristics—2001–2010 NHANES (n = 10,822) *.

	DASH score Continuous (n = 10,822)	DASH score (<4.5 vs. ≥4.5) Categorical (n = 10,822)		Allostatic load index (n = 10,822)	
	β (95% CI)	OR	95% CI	β	95% CI
**Age (years):**	0.0073 (0.0046, 0.0099)	−0.016	−0.021, −0.012	0.029	0.027, 0.030
**Sex:**					
Male	Ref.	Ref.	--	Ref.	--
Female	0.28 (0.22, 0.33)	−0.53	−0.64, −0.41	−0.49	−0.53, −0.44
**Race/Ethnicity:**					
Mexican American	Ref.	Ref.	--	Ref.	--
Other Hispanic	−0.028 (−0.17, 0.11)	0.14	−0.14, 0.42	0.026	−0.089, 0.14
Non-Hispanic White	−0.25 (−0.38, −0.12)	0.41	0.20, 0.62	−0.034	−0.12, 0.048
Non-Hispanic Black	−0.41 (−0.54, −0.29)	0.81	0.59, 1.04	0.0053	−0.069, 0.079
Other	0.096 (−0.098, 0.29)	−0.056	−0.37, 0.26	0.12	−0.029, 0.27
**Education:**					
<Less Than 9th Grade	Ref.	Ref.	--	Ref.	--
9–11th Grade	−0.29 (−0.42, −0.15)	0.45	0.18, 0.73	−0.081	−0.21, 0.048
High School Graduate/General Education Development or Equivalent	−0.31 (−0.45, −0.17)	0.44	0.19, 0.68	−0.062	−0.18, 0.061
Some College or AA degree	−0.22 (−0.36, −0.091)	0.34	0.11, 0.57	−0.092	−0.20, 0021
College Graduate or above	0.045 (−0.10, 0.19)	−0.17	−0.43, 0.088	−0.29	−0.39, −0.19
**Marital status:**					
Married/Living with partner	Ref.	Ref.	--	Ref.	--
Other	0.014 (−0.059, 0.089)	−0.032	−0.15, 0.091	0.048	−0.011, 0.11
**Poverty-income ratio:**					
<100%	Ref.	Ref.	--	Ref.	--
100%–<200%	−0.062 (−0.16, 0.033)	0.092	−0.11, 0.29	−0.053	−0.13, 0.025
≥200%	−0.11 (−0.21, −0.0077)	0.16	−0.025, 0.35	−0.14	−0.24, −0.050
**Smoking status:**					
Never smoker	Ref.	Ref.	--	Ref.	--
Ex-smoker	−0.10 (−0.19, −0.015)	0.16	−0.0017, 0.33	−0.034	−0.10, 0.038
Current smoker	−0.25 (−0.33, −0.16)	0.57	0.39, 0.75	0.28	0.21, 0.35
**Alcohol consumption** **(≥12 glasses in the past 12 months):**					
Yes	Ref.	Ref.	--	Ref.	--
No	−0.0053 (−0.095, 0.084)	−0.023	−0.16, 0.12	0.072	0.0064, 0.13
**Physical activity:**					
Yes	Ref.	Ref.	--	Ref.	--
No	−0.21 (−0.29, −0.13)	0.35	0.22, 0.47	0.035	−0.017, 0.088
**Body mass index (kg/m^2^):**	-0.018 (−0.024, −0.0121)	0.024	0.014, 0.035	0.11	0.10, 0.12
**Self-rated health:**					
Excellent/Very good/Good	Ref.	Ref.	--	Ref.	--
Fair/Poor	−0.029 (−0.11, 0.051)	0.014	−0.14, 0.17	0.14	0.066, 0.22

* Fully-adjusted model. Abbreviations: CI = confidence interval, DASH = Dietary Approaches to Stop Hypertension, NHANES = National Health and Nutrition Examination Survey, SEM = standard error of the mean.

**Table 4 nutrients-11-02311-t004:** Linear Regression Models for Dietary Approaches to Stop Hypertension Diet Score and its Component Scores as a Predictor of Allostatic Load Index Score—2001–2010 NHANES.

	Unadjusted Model (n = 11,630)			Adjusted Model * (n = 10,822)		
Linear Regression Models:	β	95% CI	P	β	95% CI	P
DASH total score (continuous)	−0.085	−0.11, −0.058	<0.0001	−0.019	−0.038, −0.00052	0.044
DASH total score (<4.5 vs. ≥4.5)	0.23	0.14, 0.31	<0.0001	0.066	0.0057, 0.13	0.032
DASH component 1 (Sat. Fat)	−0.21	−0.30, −0.11	<0.0001	−0.054	−0.12, 0.014	0.12
DASH component 2 (Tot. Fat)	−0.13	−0.21, −0.046	0.003	0.036	−0.020, 0.092	0.20
DASH component 3 (Protein)	0.079	−0.0024, 0.16	0.057	0.0016	−0.058, 0.061	0.95
DASH component 4 (Cholesterol)	−0.25	−0.34, −0.18	<0.0001	−0.049	−0.11, 0.015	0.13
DASH component 5 (Fiber)	−0.22	−0.34, −0.11	<0.0001	−0.092	−0.18, −0.0032	0.043
DASH component 6 (Magnesium)	−0.46	−0.58, −0.34	<0.0001	−0.22	0.32, −0.14	<0.0001
DASH component 7 (Calcium)	−0.12	−0.20, −0.041	0.004	−0.0049	−0.063, 0.053	0.87
DASH component 8 (Potassium)	0.55	−0.22, 1.33	0.16	0.069	−0.88, 1.02	0.88
DASH component 9 (Sodium)	−0.088	−0.42, 0.24	0.59	0.087	−0.16, 0.33	0.48

* Adjusted for demographic, socioeconomic, lifestyle and health characteristics. Abbreviations: CI = confidence interval, DASH = Dietary Approaches to Stop Hypertension, NHANES = National Health and Nutrition Examination Survey, Sat = Saturated, Tot = Total.

**Table 5 nutrients-11-02311-t005:** Cox Proportional Hazard Models for Dietary Approaches to Stop Hypertension Diet and Allostatic Load Index Scores as Predictors of All-Cause and Cause-Specific Mortality—2001–2010 NHANES * (n = 10,822).

	All-Cause Mortality (n = 10,822)		Cardiovascular Mortality (n = 10,822)		Cancer Mortality (n = 10,822)	
	Log_e_(HR)	95% CI	Log_e_(HR)	95% CI	Log_e_(HR)	95% CI
**Allostatic load index score:**	0.080	0.012, 0.14	0.074	−0.079, 0.23	0.065	−0.083, 0.21
**DASH total score (continuous):**						
Model I:						
DASH total score	−0.023	−0.076, 0.030	0.045	−0.11, 0.21	−0.052	−0.17, 0.068
Model II:						
DASH total score	−0.022	−0.075, 0.031	0.045	−0.12, 0.21	−0.051	−0.17, 0.069
Allostatic index score	0.079	0.011, 0.14	0.075	−0.079, 0.23	0.064	−0.084, 0.21
Model III:						
DASH total score	−0.087	−0.29, 0.12	−0.12	−0.56, 0.32	−0.029	−0.34, 0.28
Allostatic load index score	0.018	−0.18, 0.21	−0.087	−0.49, 0.32	0.084	−0.24, 0.41
DASH total score x Allostatic load index score	0.019	−0.035, 0.074	0.048	−0.058, 0.15	−0.0066	−0.093, 0.080
**DASH total score (<4.5 vs. ≥4.5):**						
Model I:						
DASH total score < 4.5	0.063	−0.11, 0.23	−0.14	−0.52, 0.24	0.019	−0.34, 0.38
Model II:						
DASH total score < 4.5	0.063	−0.10, 0.23	−0.14	−0.52, 0.24	0.017	−0.35, 0.38
Allostatic index score	0.080	0.012, 0.14	0.074	−0.079, 0.22	0.065	−0.083, 0.21
Model III:						
DASH total score < 4.5	0.43	−0.17, 1.04	0.48	−0.49, 1.47	0.13	−0.86, 1.12
Allostatic load index score	0.16	0.014, 0.32	.21	−0.0042, 0.42	0.091	−0.18, 0.37
DASH total score < 4.5 x Allostatic load index score	−0.11	−0.27, 0.061	−0.18	−0.42, 0.060	−0.033	−0.32, 0.25
**DASH Component 1 (Sat. Fat):**						
Model I:						
DASH Component 1	−0.13	−0.35, 0.086	0.046	−0.54, 0.63	−0.34	−0.87, 0.18
Model II:						
DASH Component 1	−0.12	−0.34, 0.088	0.048	−0.54, 0.63	−0.34	−0.86, 0.18
Allostatic load index score	0.079	0.012, 0.14	0.074	−0.079, 0.23	0.064	−0.084, 0.21
Model III:						
DASH Component 1	−0.20	−0.89, 0.48	−0.95	−2.58, 0.68	0.24	−1.15, 1.64
Allostatic load index score	0.072	−0.039, 0.18	−0.041	−0.29, 0.21	0.12	−0.12, 0.36
DASH Component 1 x Allostatic load index score	0.022	-0.15, 0.19	0.28	-0.11, 0.68	-0.17	-.55, 0.20
**DASH Component 2 (Tot. Fat):**						
Model I:						
DASH Component 2	0.081	−0.077, 0.24	0.029	−0.36, 0.42	0.012	−0.33, 0.36
Model II:						
DASH Component 2	0.077	−0.078, 0.23	0.026	−0.37, 0.42	0.0092	−0.33, 0.35
Allostatic load index score	0.079	0.013, 0.15	0.074	−0.079, 0.23	0.065	−0.082, 0.21
Model III:						
DASH Component 2	−0.031	−0.59, 0.53	−0.092	−1.19, 1.011	0.26	−0.72, 1.25
Allostatic load index score	0.064	−0.042, 0.17	0.058	−0.17, 0.29	0.099	−0.11, 0.31
DASH Component 2 x Allostatic load index score	0.032	−0.12, 0.18	0.033	−0.24, 0.30	−0.075	−0.35, 0.20
**DASH Component 3 (Protein):**						
Model I:						
DASH Component 3	−0.17	−0.34, −0.010	−0.008	−0.41, 0.39	-0.079	−0.42, 0.27
Model II:						
DASH Component 3	−0.17	−0.34, −0.012	−0.012	−0.41, 0.39	−0.081	−0.43, 0.26
Allostatic load index score	0.081	0.013, 0.15	0.074	−0.079, 0.23	0.065	−0.083, 0.21
Model III:						
DASH Component 3	−0.44	−0.98, 0.082	−0.60	−1.63, 0.42	−0.044	−1.09, 1.01
Allostatic load score	0.058	−0.012, 0.13	0.019	−0.15, 0.19	0.068	−0.088, 0.22
DASH Component 3 x Allostatic load index score	0.077	−0.058, 0.21	0.17	−0.11, 0.44	−0.011	−0.31, 0.28
**DASH Component 4 (Cholesterol):**						
Model I:						
DASH Component 4	−0.0203	−0.21, 0.17	0.028	−0.42, 0.47	−0.18	−0.57, 0.19
Model II:						
DASH Component 4	−0.016	−0.21, 0.18	0.031	−0.41, 0.47	−0.18	−0.57, 0.20
Allostatic load index score	0.080	0.012, 0.15	0.074	−0.079, 0.22	0.064	−0.084, 0.21
Model III:						
DASH Component 4	0.059	−0.45, 0.57	0.53	−0.59, 1.66	−0.66	−1.59, 0.28
Allostatic load score	0.088	-0.0018, 0.18	0.12	-0.060, 0.31	0.018	-0.18, 0.21
DASH Component 4 x Allostatic load index score	−0.022	−0.15, 0.11	−0.14	−0.42, 0.13	0.14	−0.12, 0.39
**DASH Component 5 (Fiber):**						
Model I:						
DASH Component 5	−0.11	−0.35, 0.13	0.44	−0.041, 0.93	−0.46	−0.92, 0.0043
Model II:						
DASH Component 5	−0.11	−0.35, 0.13	0.44	−0.043, 0.93	−0.45	−0.92, 0.0071
Allostatic load index score	0.080	0.013, 0.15	0.073	−0.079, 0.23	0.063	−0.084, 0.21
Model III:						
DASH total score	−0.17	−0.89, 0.55	0.79	−0.56, 2.15	−0.58	−1.78, 0.61
Allostatic load score	0.077	0.00021, 0.15	0.098	−0.070, 0.26	0.058	−0.099, 0.22
DASH Component 5 x Allostatic load index score	0.018	−0.16, 0.20	−0.10	−0.45, 0.25	0.039	−0.27, 0.36
**DASH Component 6 (Magnesium):**						
Model I:						
DASH Component 6	−0.093	−0.35, 0.16	0.37	−0.27, 1.02	−0.047	−0.58, 0.49
Model II:						
DASH total score	−0.075	−0.33, 0.19	0.39	−0.26, 1.04	−0.031	−0.57, 0.51
Allostatic load index score	0.079	0.010, 0.14	0.079	−0.076, 0.23	0.064	−0.082, 0.21
Model III:						
DASH Component 6	−0.72	−1.58, 0.14	−0.63	−2.38, 1.11	−0.30	−1.80, 1.20
Allostatic load score	0.046	−0.031, 0.12	0.016	−0.16, 0.19	0.050	−0.12, 0.22
DASH Component 6 x Allostatic load index score	0.19	−0.049, 0.43	0.29	−0.13, 0.73	0.082	−0.35, 0.51
**DASH Component 7 (Calcium):**						
Model I:						
DASH Component 7	0.084	−0.10, 0.27	−0.12	−0.64, 0.38	0.17	−0.28, 0.63
Model II:						
DASH Component 7	0.086	−0.098, 0.27	−0.13	−0.64, 0.38	0.17	−0.28, 0.63
Allostatic load score	0.080	0.013, 0.15	0.074	−0.080, 0.23	0.066	−0.083, 0.21
Model III:						
DASH Component 7	0.087	−0.54, 0.72	−0.71	−2.15, 0.73	0.28	−0.98, 1.55
Allostatic load index score	0.081	-0.0066, 0.17	0.019	-0.18, 0.22	0.077	-0.083, 0.24
DASH Component 7 x Allostatic load index score	−0.00027	−0.16, 0.16	0.16	−0.17, 0.50	−0.032	−0.35, 0.28
**DASH Component 8 (Potassium):**						
Model I:						
DASH Component 8	1.20	0.45, 1.96	−65.78	−93.30, −38.26	2.22	1.32, 3.13
Model II:						
DASH Component 8	1.21	0.46, 1.97	−71.66	−859.75, 716.42	2.23	1.32, 3.14
Allostatic load index score	0.080	0.012, 0.15	0.074	−0.078, 0.23	0.065	−0.082, 0.21
Model III:						
DASH Component 8	−1.97	−14.75, 10.79	−62.64	−77.01, −48.28	5.44	1.06, 9.83
Allostatic load index score	0.079	0.011, 0.15	0.074	−0.078, 0.23	0.068	−0.079, 0.22
DASH Component 8 x Allostatic load index score	0.89	−2.55, 4.34	−0.62	−1.97, 0.73	−0.98	−2.31, 0.33
**DASH Component 9 (Sodium):**						
Model I:						
DASH Component 9	−0.25	−0.79, 0.29	70.31	--	−0.16	−1.36, 1.02
Model II:						
DASH Component 9	−0.27	−0.80, 0.26	66.16	--	−0.18	−1.36, 1.00
Allostatic load index score	0.080	0.013, 0.15	0.072	−0.081, 0.22	0.065	−0.082, 0.21
Model III:						
DASH Component 9	1.04	−2.10, 4.19	--	--	−0.34	−3.90, 3.21
Allostatic load index score	0.44	−0.37, 1.26	--	--	0.016	−0.99, 1.02
DASH Component 9 x Allostatic load index score	−0.37	−1.21, 0.46	--	--	0.049	−0.97, 1.07

* All models are adjusted for demographic, socioeconomic, lifestyle and health characteristics. Abbreviations: CI = confidence interval, HR=hazard ratio, DASH = Dietary Approaches to Stop Hypertension, NHANES = National Health and Nutrition Examination Survey.

**Table 6 nutrients-11-02311-t006:** Direct, Indirect and Total Effects for Mediation of Dietary Approaches to Stop Hypertension Diet-Mortality Relationships by Allostatic Load Index—2001–2010 NHANES *.

	Direct Effect		Indirect Effect		Total Effect	
	Estimate	95% CI	Estimate	95% CI	Estimate	95% CI
**All-Cause Mortality:**						
DASH total score (continuous)	0.94	0.92, 0.97	0.99	0.99, 1.00	0.95	0.92, 0.97
DASH total score (<4.5 vs. ≥4.5)	1.24	1.14, 1.36	1.00	0.99, 1.00	1.24	1.14, 1.36
DASH component 1 (Sat. Fat)	0.77	0.69, 0.87	0.99	0.99, 1.00	0.77	0.69, 0.87
DASH component 2 (Tot. Fat)	1.07	0.99, 1.17	0.99	0.99, 1.00	1.07	0.99, 1.17
DASH component 3 (Protein)	0.79	0.73, 0.86	1.00	0.99, 1.00	0.79	0.73, 0.86
DASH component 4 (Cholesterol)	1.03	0.95, 1.13	0.99	0.99, 1.00	1.03	0.94, 1.13
DASH component 5 (Fiber)	0.84	0.74, 0.94	0.99	0.98, 1.00	0.84	0.75, 0.94
DASH component 6 (Magnesium)	0.95	0.84, 1.08	0.99	0.99, 1.00	0.95	0.84, 1.07
DASH component 7 (Calcium)	0.88	0.80, 0.96	0.99	0.99, 1.00	0.88	0.80, 0.97
DASH component 8 (Potassium)	4.63	1.79, 11.95	1.01	0.99, 1.02	4.67	1.81, 12.04
DASH component 9 (Sodium)	0.56	0.40, 0.79	0.99	0.99, 1.00	0.56	0.40, 0.79
**Cardiovascular disease-specific Mortality:**						
DASH total score (continuous)	1.06	0.94, 1.21	0.99	0.98, 0.99	1.06	0.93, 1.20
DASH total score (<4.5 vs. ≥4.5)	0.84	0.57, 1.25	1.02	1.00, 1.04	0.86	0.58, 1.28
DASH component 1 (Sat. Fat)	1.27	0.74, 2.20	0.97	0.95, 0.99	1.24	0.73, 2.14
DASH component 2 (Tot. Fat)	1.23	0.82, 1.85	0.98	0.97, 0.99	1.21	0.81, 1.83
DASH component 3 (Protein)	0.78	0.52, 1.16	1.01	0.99, 1.01	0.79	0.53, 1.18
DASH component 4 (Cholesterol)	1.18	0.77, 1.79	0.98	0.96, 0.99	1.15	0.76, 1.75
DASH component 5 (Fiber)	1.22	0.72, 2.06	0.96	0.94, 0.99	1.17	0.69, 1.98
DASH component 6 (Magnesium)	1.48	0.84, 2.61	0.96	0.92, 0.99	1.42	0.80, 2.50
DASH component 7 (Calcium)	1.06	0.68, 1.67	0.99	0.99, 1.00	1.07	0.68, 1.67
DASH component 8 (Potassium)	--	--	--	--	--	--
DASH component 9 (Sodium)	--	--	--	--	--	--
**Cancer-specific Mortality:**						
DASH total score (continuous)	0.92	0.82, 1.02	1.00	0.99, 1.00	0.92	0.82, 1.03
DASH total score (<4.5 vs. ≥4.5)	1.13	0.79, 1.61	0.99	0.98, 1.02	1.13	.79, 1.61
DASH component 1 (Sat. Fat)	0.67	0.42, 1.06	1.00	0.98, 1.02	0.67	0.43, 1.06
DASH component 2 (Tot. Fat)	0.92	0.65, 1.28	1.00	0.99, 1.01	0.92	0.66, 1.28
DASH component 3 (Protein)	1.10	0.80, 1.51	0.99	0.99, 1.01	1.10	0.80, 1.51
DASH component 4 (Cholesterol)	0.68	0.47, 1.00	1.00	0.98, 1.02	0.69	0.47, 1.00
DASH component 5 (Fiber)	0.65	0.39, 1.06	1.00	0.98, 1.03	0.65	0.39, 1.07
DASH component 6 (Magnesium)	0.99	0.60, 1.65	1.00	.97, 1.04	0.99	0.60, 1.65
DASH component 7 (Calcium)	0.95	0.65, 1.38	1.00	0.99, 1.00	0.95	0.65, 1.39
DASH component 8 (Potassium)	2.04	0.043, 96.77	0.99	0.94, 1.06	2.03	0.043, 96.41
DASH component 9 (Sodium)	1.02	0.23, 4.57	1.00	0.98, 1.01	1.02	0.22, 4.58

* All models are adjusted for demographic, socioeconomic, lifestyle and health characteristics. Abbreviations: CI = confidence interval, DASH = Dietary Approaches to Stop Hypertension, NHANES = National Health and Nutrition Examination Survey.

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
