# Peer review of "Mediating-Moderating Effect of Allostatic Load on the Association between Dietary Approaches to Stop Hypertension Diet and All-Cause and Cause-Specific Mortality: 2001–2010 National Health and Nutrition Examination Surveys"

_nutrients, 2019, doi:10.3390/nu11102311_

Round 1
Reviewer 1 Report
The authors conducted extensive analyses in examining mediation and moderation relationships between DASH and Allostatic load but a lot can be improved in the presentation and text of the manuscript to improve readability and sustain reader's interest throughout the paper.
Suggestions/comments as follows:
Abtract - The authors used a modified DASH score but did not mention this in the abstract. They present only a mean value of a non-standard score without any reference. A median (range) might be better presentation for the population, especially because the range is totally different from the standard DASH score of 8-40. Abstract - The same should be addressed for the allostatic load score. Introduction - This section can be tightened to make it shorter especially describing DASH pattern, which has been well described in previous reports. It is up to the authors' judgement as they see the need for the target audience to understand the background. Materials and Methods - Here some detail about the well known NHANES project can be shortened - again left to the discretion of the authors Materials and Methods - The exclusion of CRP >10mg/dl should be explained. Is it an implausible value? What is the normal range? Is the intention to exclude people with known inflammatory diseases/conditions at baseline? Materials and Methods - DASH score used in this study is a modified one. Even if Mellen Index is a well known modification, the section on describing the DASH score needs a table for better readability and understanding of this section. Materials and Methods - The section on description of Allostatic load also needs a table. The above tables could be added as supplementary tables if the limit does not permit in the main text. Results - Restate the observation in Line 251 - Non-alcoholic drinkers (<12 vs. ≥12 glasses in the past 12 months). This detail will help to understand the reference again although it is in the Table 1 Table 1 - Does population distribution by age categories provide any specific information in addition to the mean (se)? Tables 2&3 - can be merged into one Discussion - In this paper, the discussion should be geared more towards the mediation and moderation between DASH and ALI. Discussing previous findings about DASH alone should not be the intent here. Please add some discussion on what the mediation findings mean and how do these findings in a larger public health context. Limitations - were addressed well. Since the authors performed extensive multiple analyses, was there any attempt to mitigate or correct for multiple testing? If yes, please mention that.Author Response
Reviewer #1 comments:
The authors conducted extensive analyses in examining mediation and moderation relationships between DASH and Allostatic load but a lot can be improved in the presentation and text of the manuscript to improve readability and sustain reader's interest throughout the paper.
Abstract- The authors used a modified DASH score but did not mention this in the abstract. They present only a mean value of a non-standard score without any reference. A median (range) might be better presentation for the population, especially because the range is totally different from the standard DASH score of 8-40.
Response to Comment #1: We acknowledge the reviewer’s comment. In the revised version of the manuscript, we have mentioned that we used a modified DASH score that is based on nutrients rather than food groups. Since this is a non-standard DASH score, we referred to it as the Mellen Index and we described it using median (range) rather than mean alone, as previously described. See track-changes within the revised manuscript, including the abstract and Results section.
Abstract - The same should be addressed for the allostatic load score.
Response to Comment #2: We acknowledge the reviewer’s comment. In the revised version of the manuscript, we addressed the same issue for the allostatic load score. Specifically, we referred to the method used to generate the allostatic load score as the Rodriqez method and we described the allostatic load score using median (range) instead of mean alone. See track-changes within the revised manuscript, including the abstract and Results section.
Introduction - This section can be tightened to make it shorter especially describing DASH pattern, which has been well described in previous reports. It is up to the authors' judgement as they see the need for the target audience to understand the background.
Response to Comment #3: We acknowledge the reviewer’s comment. In the revised version of the manuscript, we have tightened the introductory section to make it shorter when describing the DASH pattern since it has been well described in previous reports. See track-changes in the Introduction section.
Materials and Methods -Here some detail about the well-known NHANES project can be shortened - again left to the discretion of the authors
Response to Comment #4: We acknowledge the reviewer’s comment. In the revised version of the manuscript, we have shortened detail about the NHANES project since it is well-known. See track-changes in the Materials and Methods section.
Materials and Methods - The exclusion of CRP >10mg/dl should be explained. Is it an implausible value? What is the normal range? Is the intention to exclude people with known inflammatory diseases/conditions at baseline?
Response to Comment #5: We acknowledge the reviewer’s comment. In the revised version of the manuscript, we explained the exclusion of CRP >10mg/dl, as follows: “The exclusion of NHANES participants with CRP >10mg/dl was intended to reduce confounding related to the presence of inflammatory diseases that may affect dietary patterns as well as mortality risks”. See track-changes in the Materials and Methods section.
Materials and Methods - DASH score used in this study is a modified one. Even if Mellen Index is a well-known modification, the section on describing the DASH score needs a table for better readability and understanding of this section.
Response to Comment #6: We acknowledge the reviewer’s comment. In the revised version of the manuscript, we added an Appendix table that clearly presents the Mellen Index in order to improve the readability and understanding of the Materials and Methods section where the modified DASH score was described. See track-changes in the Materials and Methods section as well as the added Appendix table.
Materials and Methods - The section on description of Allostatic load also needs a table. The above tables could be added as supplementary tables if the limit does not permit in the main text.
Response to Comment #7: We acknowledge the reviewer’s comment. In the revised version of the manuscript, we added an Appendix table that clearly presents the Allostatic Load Index as described in the text based on the Rodriqez and colleagues. See track-changes in the Materials and Methods section.
Results - Restate the observation in Line 251 - Non-alcoholic drinkers (<12 vs. ≥12 glasses in the past 12 months). This detail will help to understand the reference again although it is in the Table 1.
Response to Comment #8: We acknowledge the reviewer’s comment. In the revised version of the manuscript, we restated the observation in Line 251 as “Non-alcoholic drinkers (<12 vs. ≥12 glasses in the past 12 months)” as described in Table 1 as well as in the Materials and Methods section. See track-changes in the Results section.
Table 1 -Does population distribution by age categories provide any specific information in addition to the mean (se)?
Response to Comment #9: We acknowledge the reviewer’s comment. Originally, we provided age categories for descriptive purposes only. However, most analyses within this manuscript are focused on age defined as a continuous variable rather than age defined as a categorical variable. In the revised version of the manuscript, we deleted the reference to age categories from the Materials and Methods section (See track-changes) as well as Table 1 (See track-changes).
Tables 2&3 -can be merged into one.
Response to Comment #10: We acknowledge the reviewer’s comment. In the revised version of the manuscript, we merged Tables 2 and 3 into one Table and made revisions throughout the manuscript accordingly (See track-changes).
Discussion -In this paper, the discussion should be geared more towards the mediation and moderation between DASH and ALI. Discussing previous findings about DASH alone should not be the intent here. Please add some discussion on what the mediation findings mean and how do these findings in a larger public health context.
Response to Comment #11: We acknowledge the reviewer’s comment. In the revised version of the manuscript, we added a paragraph to discuss mediation and moderation between DASH and ALI, and specifically what the mediation findings mean and how do these findings in a larger public health context (See track-changes).
Limitations -were addressed well. Since the authors performed extensive multiple analyses, was there any attempt to mitigate or correct for multiple testing? If yes, please mention that.
Response to Comment #12: We acknowledge the reviewer’s comment. In the revised version of the manuscript, we addressed the issue of multiple-testing within the Discussion section, as follows:
Of note, with ten exposure variables (DASH total score and nine component scores) and three outcome variables (all-cause, cardiovascular-specific and cancer-specific mortality rates), similar hypotheses were tested 30 times, potentially reducing alpha to 0.002 per hypothesis after Bonferroni correction.

Reviewer 2 Report
This study examined mediating-moderating effects of allostatic load score on association between DASH diet and all-cause and cause-specific mortality risks among 21 11,630 adults in the United States using national survey data.
Abstract:
The authors claimed that this was a retrospective cohort study. However, according to the methodology and dataset used, I would say that this was a secondary analysis of national survey.
Lack of implication information in abstract.
Introduction:
Background information on allostatic load should be strengthened in the introduction. It would be helpful if existing literature on using allostatic load on nutrition studies were summarized in the introduction.
Methods:
On P3, imputation process needed more detailed description. It was noted that more than 20% data has been imputated. This might need to discuss in the Discussion section.
The authors did not explain well why they calculate DASH score based on nutrients rather than food group.
Findings:
In Table 1, n was not consistent among different column. I would suggest calculating mean and percentage based on n=10,822.
In all tables, p values were not reported.
Discussion:
Discussion is related to findings and current literature. The discussion on the implications of findings was not sufficient.
Author Response
Reviewer #2 comments:
This study examined mediating-moderating effects of allostatic load score on association between DASH diet and all-cause and cause-specific mortality risks among 21 11,630 adults in the United States using national survey data.
Abstract: The authors claimed that this was a retrospective cohort study. However, according to the methodology and dataset used, I would say that this was a secondary analysis of national survey. Lack of implication information in abstract.
Response to Comment #1: We acknowledge the reviewer’s comment. In the revised version of the manuscript, we have substituted the phrase “retrospective cohort study” with “secondary analysis of survey data”, throughout the manuscript including the Abstract, Introduction and Discussion sections. We also added implication information in the abstract (See track-changes).
Introduction: Background information on allostatic load should be strengthened in the introduction. It would be helpful if existing literature on using allostatic load on nutrition studies were summarized in the introduction.
Response to Comment #2: We acknowledge the reviewer’s comment. In the revised version of the manuscript, we have strengthened the background information on allostatic load and provided a paragraph that further links diet to allostatic load (See track-changes).
Methods: On Page 3, imputation process needed more detailed description. It was noted that more than 20% data has been imputated. This might need to discuss in the Discussion section. The authors did not explain well why they calculate DASH score based on nutrients rather than food group.
Response to Comment #3: We acknowledge the reviewer’s comment. In the revised version of the manuscript, we had added information regarding the imputation process within the Materials and Methods section. We also provided additional information in the Discussion section. We also explained within the Materials and Methods section the reason for calculating the DASH score based on nutrients rather than food groups, as follows:
Since the purpose of this study was to elucidate the biopsychosocial mechanism underlying the diet-mortality link and for consistency with previously conducted studies using NHANES, we operationalized the DASH dietary pattern using a methodology developed by Mellen and colleagues.
See track-changes.
Findings: In Table 1, n was not consistent among different columns. I would suggest calculating mean and percentage based on n=10,822. In all tables, p values were not reported.
Response to Comment #4: We acknowledge the reviewer’s comment. However, the final study sample consists of 11,630 NHANES participants, of whom 10,822 NHANES participants were included in multivariable analyses. Since 11,630 NHANES participants were analyzed in the context of univariate and bivariate analyses, we created a new Table 1 for baseline characteristics and the old Table 1 is now Table 2 which includes Cox regression models. In addition, given the epidemiologic nature of this study, we decided to present 95% confidence intervals rather than p-values. See track changes in the Results section.
Discussion: Discussion is related to findings and current literature. The discussion on the implications of findings was not sufficient.
Response to Comment #5: We acknowledge the reviewer’s comment. In the revised version of the manuscript, we added a paragraph in the Discussion section regarding implications of study findings, especially with regard to mediation (See track-changes).
